# A Shortage of FTH Induces ROS and Sensitizes RAS-Proficient Neuroblastoma N2A Cells to Ferroptosis

**DOI:** 10.3390/ijms22168898

**Published:** 2021-08-18

**Authors:** Ruiqing Lu, Yinan Jiang, Xianxin Lai, Shujie Liu, Litao Sun, Zhong-Wei Zhou

**Affiliations:** 1School of Medicine, Sun Yat-Sen University, Shenzhen 518107, China; lurq@m.scnu.edu.cn (R.L.); jiangyn23@mail2.sysu.edu.cn (Y.J.); laixx7@mail2.sysu.edu.cn (X.L.); liushj53@mail2.sysu.edu.cn (S.L.); 2School of Public Health (Shenzhen), Sun Yat-Sen University, Shenzhen 518107, China

**Keywords:** FTH, ROS, ferroptosis, neuroblastoma

## Abstract

Ferroptosis, an iron-dependent form of programmed cell death, has excellent potential as an anti-cancer therapeutic strategy in different types of tumors, especially in RAS-mutated ones. However, the function of ferroptosis for inhibiting neuroblastoma, a common child malignant tumor with minimal treatment, is unclear. This study investigated the anti-cancer function of ferroptosis inducer Erastin or RSL3 in neuroblastoma N2A cells. Our results show that Erastin or RSL3 induces ROS level and cell death and, therefore, reduces the viability of RAS-proficient N2A cells. Importantly, inhibitors to ferroptosis, but not apoptosis, ameliorate the high ROS level and viability defect in Erastin- or RSL3-treated cells. In addition, our data also show that N2A cells are much more sensitive to ferroptosis inducers than primary mouse cortical neural stem cells (NSCs) or neurons. Moreover, a higher level of ROS and PARylation is evidenced in N2A, but not NSCs. Mechanically, ferritin heavy chain 1 (*Fth*), the ferroxidase function to oxidate redox-active Fe^2+^ to redox-inactive Fe^3+^, is likely responsible for the hypersensitivity of N2A to ferroptosis induction since its expression is lower in N2A compared to NSCs; ectopic expression of *Fth* reduces ROS levels and cell death, and induces expression of GPX4 and cell viability in N2A cells. Most importantly, neuroblastoma cell lines express a significantly low level of *Fth* than almost all other types of cancer cell lines. All these data suggest that Erastin or RSL3 induce ferroptosis cell death in neuroblastoma N2A cells, but not normal neural cells, regardless of RAS mutations, due to inadequate FTH. This study, therefore, provides new evidence that ferroptosis could be a promising therapeutic target for neuroblastoma.

## 1. Introduction

Neuroblastoma is a common extracranial solid tumor in children, accounting for approximately 15% of childhood cancer deaths, with a five-year survival rate of about 75%, and a 50–60% recurrence rate at a high-risk grade [1,2]; it can occur in the neck, chest, abdomen, and even in the bone marrow. Chemotherapy drugs, such as cyclophosphamide, vincristine, amycin, cisplatin, etoposide, and temozolomide, are frequently used for neuroblastoma treatment [3]. However, drug resistance and toxic side effects often occur clinically, such as bone marrow failure [4,5].

The primary mechanism of chemotherapy, at present, is to induce tumor cell death [6,7]. Ferroptosis, a non-apoptotic form of cell death induced by an iron-dependent lipid peroxide accumulation, is characterized mainly by cell volume shrinkage and increased mitochondrial membrane density without typical features of apoptosis, e.g., coagulation of chromosomes, formation of apoptosis bodies, and production of cleaved caspase-3 [8]. Since its discovery in 2012 [8], ferroptosis has attracted the attention of scientists from different fields and is involved in neurological diseases, heart diseases, liver diseases, kidney diseases, gastrointestinal diseases [9]. Erastin and RSL3, inducers of ferroptosis, were identified and proved to have a synergetic lethal effect in engineered human cells harboring RAS mutations [10,11], an oncogenic mutation often found in different types of cancers [12]. Ferroptosis has then become a key tumor suppression mechanism [13,14]. However, the role of ferroptosis in neuroblastoma is not well studied.

Ferroptosis is an iron- and ROS-dependent form of programmed cell death (PCD). Iron is essential for life; however, excessive iron is harmful to the cells or organisms since it can directly catalyze free radical formation via Fenton reactions to induce reactive oxygen species (ROS), especially lipid peroxidation (and, therefore, cell death) [15,16,17]. Intracellular iron homeostasis is controlled by a series of genes, including transferrin (TFR) and divalent metal transporter 1 (DMT1), which transport iron into the cells, while ferroportin (FPN) is responsible for iron export [18]. Free iron is stored by ferritin, a complex composed of ferritin heavy chain 1(FTH1, or FTH) and ferritin light chain (FTL), to protect cells from oxidative stress [19,20]. Whereas FTL functions as a scaffold to support the ferritin multimeric complex, FTH plays a role in converting redox-active Fe^2+^ to redox-inactive Fe^3+^ [20]. Defecting FTH has been shown to induce ferroptosis and cause cardiomyopathy, which can be effectively reversed by membrane protein Slc7a11 in myocardial cells. On the other hand, expression of FTH decreased in ferroptosis-sensitive cells with Ras mutation as compared to ferroptosis-resistant cells [10], indicating the importance of FTH in suppressing ferroptosis cell death. In addition to iron and iron-mediated ROS production, reduced glutathione (GSH) is also a central player in ferroptosis [21]. GSH is required for eliminating oxidized phospholipids by glutathione peroxidase 4 (GPX4), which functions to convert GSH to oxidized glutathione (GSSG) while reducing lipid hydroperoxides [21]. Either deletion of GSH or inhibition of GPX4 can induce ferroptosis. Therefore, ferroptosis is a GPX4-regulated cell death that could be blocked by iron chelates as well as lipid antioxidants [22]. Additionally, ferroptosis suppressor protein 1 (FSP1), also called apoptosis-inducing factor mitochondria-associated 2 (AIFM2), works together with the CoQ10-NAD(P)H pathway to suppress lipid peroxidation and, thereby, ferroptosis, in parallel with GPX4 [23,24]. Whether (and how) ferroptosis-associated molecules play a role in neuroblastoma suppression is unknown.

A large amount of ROS is produced during ferroptosis due to breakage of intracellular redox steady [8]. ROS could cause lipid peroxidation or DNA damage, which can further activate poly(ADP-ribosome) enzyme 1 (PARP1), and cause poly(ADP-ribose) (PAR) formation [25,26]. PAR is a type of protein post-translation modification produced by PARPs enzymes and plays an important role in cell stress response, DNA damage repair, transcription regulation, and metabolic regulation [25]. Interestingly, it has been reported that PARP1 regulates the expression of GPX4 [27,28], the key factor of ferroptosis, and the transcriptional activity of Nrf2 [29], another critical regulator of ferroptosis. Moreover, upon oxidative stress, GPX4 promotes cell death by regulating the nucleus relocation of apoptosis-inducing factor AIF [30], a core member of the PARP1–dependent cell death signaling pathway [31]. However, the function of PARP1 and PARP1-mediated PAR in ferroptosis is unclear.

This paper compared the effects of ferroptosis inducers on cell viability between mouse neuroblastoma N2A cells and primary cortical neural stem cells or neurons. The mechanism that sensitizes N2A to ferroptosis and the role of PARP1-mediated PAR during ferroptosis are also investigated.

## 2. Results

### 2.1. Mouse Neuroblastoma N2A Cells Are Sensitive to Erastin and RSL3

To explore the effects of Erastin and RLS3 on neuroblastoma cells, we treated mouse Neuro 2A (N2a) cells with 10 μM of Erastin or 5 μM of RSL3. Bright-field microscopic imaging shows an apparent change of cellular morphology 12 h after incubation with either Erastin or RSL3 compared to DMSO control (Figure 1A). Quantification results show a significant reduction of both total length and number of primary neurites (Figure 1B), confirming an alteration of cell morphology after Erastin or RSL3 treatment. Then cell viability was analyzed by CCK8 assay, which showed that cellular viability was dramatically reduced 24 h after Erastin- or RSL3-treatment compared to DMSO control (Figure 1C), even with a concentration as low as 2 μM of Erastin or 1 μM of RSL3. To further test the sensitivity of N2A cells to ferroptosis, we titter lowered the concentration of RSL3 and found that 0.75 μM was significantly sufficient to inhibit the viability of N2A cells, even at 8 h after treatment (Figure 1D). All of these suggest that Erastin or RSL3 treatment affects both the morphology and viability of N2A cells.

### 2.2. Erastin or RSL3 Treatment Leads to Cell Death without Affecting Proliferation

To investigate how Erastin and RSL3 affected cell viability, we first analyzed the cell proliferation by BrdU plus labeling. For this, cells were incubated with Erastin or RSL3 for 23 h and then BrdU for 1 h before fixation. Cells were then stained with an antibody against BrdU (Figure 2A). Our data show that neither Erastin nor RSL3 significantly affects the cell population incorporated with BrdU (Figure 2B), suggesting that these drugs do not alter cellular proliferation in N2A cells.

To further explore the cause of cell viability impairment upon Erastin or RSL3 treatment, N2A cells were stained with propidium iodide (PI) for microscopic analysis 12 h after treatment of Erastin or RSL3 (Figure 2C). A large amount of PI-positive (PI+) cells were observed following Erastin or RSL3 treatment, which increased PI-reacted cells, even with 2 μM and 1 μM, respectively (Figure 2C,D), indicating that incubation of these drugs caused N2A cell death. Moreover, cell death was induced with as low as 0.5 μM of RSL3 in 6 h of treatment (Appendix A). To further confirm cell death phenotype, N2A cells were stained with crystal violet and visualized under a light microscope. As shown in Figure 2E, cell survival was greatly affected even with low concentrations, 2 μM and 1 μM for Erastin and RSL3, respectively (Figure 2E), and few cells remained when the drug concentration increased to 10 μM for Erastin and 5 μM for RSL3. These data indicate that N2A cells are sensitive to Erastin and RSL3.

### 2.3. Erastin and RSL3 Induce ROS

To study the cause of cell death induced by Erastin and RSL3, we then examined ROS levels in Erastin- or RSL3-treated N2A cells since both Erastin and RSL3 are commonly used as inducers of ferroptosis, usually accompanied by upregulation of reactive oxygen species (ROS)-induced lipid peroxidation. N2A cells were penetrated with small fat-soluble molecule BODIPY 581/591 C11, an indicator of lipid peroxidation, and analyzed with FACS 24 h after Erastin- or RSL3-treatment. FACS data show that 10 μM Erastin could induce a double amount of lipid peroxidation levels (Figure 3A left, blue and Figure 3B) and RSL3 (5 μM), although it has less efficiency than Erastin; it also significantly elevates the lipid peroxidation level compared to the control in N2A cells (Figure 3A left, green and Figure 3B). All of these indicate that Erastin and RSL3 may lead to an accumulation of ROS and, therefore, lipid peroxidation, which then causes ferroptosis in N2A cells.

### 2.4. Ferroptosis Inhibitor Liproxstatin-1 Blocks Effect Caused by Erastin and RSL3 in N2A Cells

To confirm a ferroptosis process in N2A cells after Erastin or RSL3 treatment, we applied liproxstatin-1 (Lip-1), an inhibitor to ferroptosis, to N2A cells before adding Erastin or RSL3. FACS analysis shows that Lip-1 suppresses entirely or partially the ROS level labeled by BODIPY 581/591 C11 in RSL3-treated or Erastin-treated N2A cells, respectively (Figure 3A,B). To further investigate the contribution of ferroptosis in cell viability affected by Erastin or RSL3, Lip-1 was applied again in N2A cells, and CCK8 assays were performed. Our data show that either 5 or 10 μM of Lip-1 can invalidate completely N2A cell viability defect caused by Erastin or RSL3 (Figure 3C). Moreover, even 0.2 and 0.5 μM of Lip-1 were sufficient to reverse RSL3-induced cell viability defects (Appendix A). Furthermore, crystal violet staining confirmed the rescue of cell survival by Lip1 after Erastin or RSl3 treatment (Figure 3D). However, Z-VAD, a specific inhibitor to apoptosis, can significantly improve the cell viability affected by apoptosis trigger STS, but not Erastin or RSL3 in N2A cells (Figure 3E). Meanwhile, western blotting analysis shows Erastin or RSL3 treatment increased the expression of HO-1 and downregulated the expression of GPX4, both of which are hall markers for ferroptosis, but do not affect the expression of cleaved caspase-3, a marker for apoptosis (Figure 3F). These data suggest Erastin or RSL3 reduces the viability of N2A cells through ferroptosis rather than the apoptosis process.

### 2.5. Erastin and RSL3 Induce Ferroptosis in N2A Cells Independent of RAS Mutations

It is well known that RSL3, RAS-selective lethal (RSL) compounds [8], specifically kill cancer cells that harbor RAS mutations. We, therefore, tested whether the mouse neuroblastoma N2A, which is also sensitive to RSL3 and Erastin, contains any RAS mutations. mRNA was extracted from N2A cells and then reverse-transcribed into cDNA for PCR amplification with primers to RAS family genes (Figure 4A). The sequencing analysis data show that, although a T96>C mutation was found in the cDNA region of the *Kras* gene (Table 1), it is a synonymous mutation and, therefore, has no effect on the protein sequence itself (Figure 4B). Moreover, neither HRAS nor HRAS has any mutations (Table 1), suggesting that Erastin or RSL3 treatment leads to ferroptosis processes independent of RAS mutations in neuroblastoma N2A cells.

### 2.6. Mouse Primary Neural Stem Cells and Neurons Are Insensitive to Ferroptosis Inducers

To detect the toxicity of Erastin and RSL3 to non-cancer cells, mouse cortical neural neural stem cells (NSCs), as well as primary neurons, were isolated and cultured from the embryonic stage of E14.5 or E17.5, respectively. Interestingly, neither Erastin nor RSL3 significantly affects on the viability of neural stem cells, even with high concentrations (Erastin up to 50 μM and RSL3 up to 10 μM) (Figure 5A). A similar result was observed in primary neurons (Figure 5B). Whereas Erastin induces a higher level of lipid peroxidation in NSCs, surprisingly, RSL3 treatment reduces lipid peroxidation compared to DMSO treated control (Figure 5C). Moreover, the RSL3-triggered decrease of lipid peroxidation was evidenced even at 4 h after treatment (Appendix A). Nevertheless, neither affected the cell survival of NSC, as measured by crystal violet staining (Figure 5D). These data suggest that Erastin or RSL3 causes toxicity to mouse neuroblastoma, but not cultured primary neural stem cells or neurons.

### 2.7. N2A Cells Express a Lower Amount of Fth Compared to Neural Stem Cells

To investigate why Erastin or RSL3 is poisonous to cancer-type neuronal-like N2A cells, but not primary ones. The expression level of RAS family genes was first compared between N2A and primary neural stem cells. Quantitative PCR (qPCR) data show that *Hras* is the highest expression within the Ras genes in both cell lines (Figure 6A). Whereas *Hras* is significantly higher in neural stem cells than in N2A cells, the expression levels of the *Kras* gene are similar (Figure 6A). As a common marker gene of neuroblastoma, expression of *Nras* is only detected in N2A cells and undetected in neural stem cells (Figure 6A). However, silencing of *Nras* has a negligible (or even worsens the) effect on the viability of N2A cells after Erastin or RSL3 treatment (Figure 6B), suggesting a high level of *Nras* is not responsible for the sensitivity of N2A cells to ferroptosis induction.

We next compared the expression of the two critical suppressors of ferroptosis, *Gpx4*, and *Fsp1*, between N2A and neural stem cells. To our disappointment, neither *Fsp1* nor *Gpx4* showed any difference in these cell lines at the mRNA level (Figure 6C). Moreover, the protein level of GPX4 analyzed by western blotting was similar between N2A and neural stem cells (Figure 6D). We then expanded our examination from *Gpx4* and *Fsp1* to other vital players in ferroptosis, including transferrin receptor 1 (*Tfr1*), which mediates uptake of circulating transferrin-bound iron, *Slc7a11*, which is a cystine/glutamate transporter and functions to suppress ferroptosis, *Slc39a14*, which is a metal transporter that can promote ferroptosis, ferritin heavy chain 1 (*Fth1*/*Fth*), and light chain (*Ftl*) [32]. Excitingly, only *Fth* showed significantly lower in N2A cells compared to the neural stem cells (Figure 6E). Since FTH functions to store free iron, which can induce lipid hyperoxidation and, thereby, ferroptosis, the lipid hyperoxidation level was first compared between N2A and the neural stem cells. Indeed, a higher level of ROS, judged by BODIPY 581/591 C11, was found in N2A cells compared to the neural stem cells (Figure 6F). Moreover, N2A cells harbor a lower fluorescence of calcein-acetoxymethyl ester, which can be loaded into living cells and quenched by binding to labile iron, than the neural stem cells (Figure 6G), suggesting a high level of labile iron. These indicate that N2A cells, correlative with a lower *Fth* expression, contain a high labile iron pool (LIP) and ROS, which may sensitize the cell to ferroptosis.

### 2.8. Fth May Be Responsible for the Sensitivity of Neuroblastoma to Ferroptosis

To further investigate the role of *Fth* in N2A cells during ferroptosis, N2A cells were transfected either with empty vector expressing Flag-tag only (Flag-EV) or Flag-FTH fusion protein expression plasmid. Cells were then treated with RSL3 for cell death analysis, which showed that overexpression of FTH significantly inhibits RSL3-induced cell death judged by PI staining (Figure 7A). Consistent with a rescue of cell death, the RSL3-triggered viability defect of N2A cells was also improved by FTH expression (Figure 7B). Moreover, FTH reduced ROS level (Figure 7C) and enhanced GPX4 expression (Figure 7D) in Erastin- or RSL4-treated N2A cells. All of these suggest that the expression of FTH increases cell viability and reduces the sensitivity of N2A cells to ferroptosis inducers by affecting GPX4 and ROS levels.

To further explore the roles of FTH-mediated ferroptosis in neuroblastoma, we compared the expression of *Fth* between neuroblastoma and other cancer cell types. For that, we download the expression data of *Fth* in 1378 cancer cell lines from the “Broad DepMap Portal”, which is the most recently processed and up-to-date CCLE datasets. After excluding 16 cell lines, including 1× adrenal cancer, 1× embryonal cancer, 1× teratoma, and 13× engineered cancer lines, expression of *Fth* in 1362 cancer cell lines belonging to 29 types of tumors were analyzed. Strikingly, neuroblastoma cell lines expressed a significantly lower level of *Fth* compared to most of the other types of cancer cell lines, except lymphoma, myeloma, leukemia, sarcoma, rhabdoid, and prostate cancer (Figure 7E). Additionally, we analyzed the expression of *Gpx4* and *Fsp1* in these datasets. We found that neuroblastoma expresses a similar amount of *Gpx4* as most other cancer cell lines. The expression of *Fsp1* is also similar to some other cancer cell lines (Appendix A). These data suggest that a low level of *Fth* is a crucial marker of neuroblastoma, which could sensitize neuroblastoma to ferroptosis. Consistent with this point, SH-SY5Y, a human neuroblastoma cell line, also shows sensitivity to RSL3 induced cell death (Appendix A).

### 2.9. Hyper PARP Activity Induced by Erastin or RSL3 Is Independent of N2A Cell Viability

Inhibition of PARP1, an enzyme that can catalyze Poly(ADP-ribose) (PAR) formation, and plays a role in DNA damage repair [33], was recently shown to promote ferroptosis via repressing SLC7A11 and has a synergetic effect in BRCA-proficient ovarian cancer [34]. We, therefore, investigated the role of PARP1 during the ferroptosis process in N2A cells. To that, the PAR level was firstly analyzed. Western blotting results showed that, in N2A cells, Erastin or RSL3 induced a high level of PAR antibody reaction (Figure 8A), which was utterly interrupted by PAPP1 inhibitors, veliparib (Veli) or rucaparib (Ruca) (Figure 8B), suggesting that Erastin or RSL3 treatment indeed activated PARP1 and promoted PAR formation in N2A cells.

To dissect the contribution of PAR formation during ferroptosis, we treated N2A cells with different doses of PARP1 inhibitors before adding Erastin or RSL3. CCK8 assays show the negligible effects of PARP1 inhibition on cell viability impaired by Erastin or RSL3 (Figure 8C). Moreover, PARP1 inhibitors, Veli or Ruca, could neither rectify abnormal expression of GPX4 and HO-1 (Figure 8B) nor eliminate the high levels of ROS triggered by Erastin or RSL3 (Figure 8D). Of note, only a superficial level of PAR was detected in neural stem cells, and the induction of PAR by Erastin or RSL3 was very mild, even not observable 24 h and 48 h after drug treatment (Figure 8E). This suggests that Erastin or RSL3 induced only a high level of PAR without any other effect on the cell viability in N2A cells.

## 3. Discussion

RAS is a small GTPase that plays a vital role in intracellular signal transduction, controlling normal physiological processes, such as cell differentiation, gene expression, cell cycles, membrane transport, cell proliferation, movement, and survival [35,36]. RAS can be combined with GTP or hydrolyzed GTP, switching between inactive and active states, which are controlled by guanine nucleotide exchange factors (GEFs) and GTPase-activating proteins (GAP), respectively [36]. The RAS protein in tumors is often independent of GEFs or/and is insensitive to GAP, which is continuously activated, resulting in abnormal cell growth [35,36]. About 30% of human cancers have been found to harbor at least one point mutation of RAS that causes chronic activation of its activity [35,36]. Therefore, RAS is a promising target for tumor therapy. Stockwell and colleagues found, during a drug screening, that Erastin and RSL3 selectively kill cells with RAS mutation through a pathway of cell death later known as ferroptosis [10]. This study found that mouse neuroblastoma N2A cells were susceptible to Erastin and RSL3 (Figure 1B,C). Moreover, our result shows that Erastin or RSL3 treatment, rather than affect the proliferation ability, leads to substantial N2A cell death (Figure 2), accompanied by ROS increase (Figure 3). Ferroptosis inhibitor Lip-1, but not the apoptotic inhibitor Z-VAD, significantly reversed the effect caused by Erastin or RSL3 (Figure 3). All of these confirm a ferroptosis induction in N2A cells. However, our sequencing result reveals that N2A cells do not contain RAS mutations (Figure 4). Although a high level of Nras was evidenced after comparing the expression of Ras genes between N2A and normal primary neural stem cells, Nras does not seem to be responsible for the hypersensitivity of N2A to ferroptosis inducers because knockdown of Nras does not abate impairment of cell viability caused by these drugs (Figure 6B). We, therefore, conclude that Erastin or RSL3 triggers the ferroptosis process in the RAS-proficient N2A cells.

N2A cells and primary neural stem cells or neurons behave differently in response to ferroptosis inducer, Erastin, or RSL3. Although we have not tested lower concentrations than those used in this study, 2 μM of Erastin or 1 μM of RSL3 is sufficient to induce N2A cell death and reduce viability nearly 50% (Figure 1 and Figure 2). However, a significant change in cell viability could neither be induced by 50 μM of Erastin (25 times lower) nor 10 μM of RSL3 (10 times lower) in neural stem cells or neurons (Figure 5). Although their enzyme activity had not been examined, these differences seem independent of Gpx4 and Fsp1 since their expression levels are similar between N2A and neural stem cells (Figure 6C,D). FTH, the subunit of ferritin, a shell protein complex consisting of 24 subunits that sequesters iron in its core in a soluble and non-toxic state, thereby alleviating the ROS mediated by iron [37], may respond to the sensitivity of N2A cells to ferroptosis treatment. The basic level of *Fth* is more than three times lower in N2A cells compared to neural stem cells. A low level of *Fth* could lead to a high level of toxic free iron and cause various modifications to DNA bases, enhancing lipid peroxidation [38]. Indeed, we found a high level of labile iron, ROS, and PAR, which is a marker for DNA damage in N2A cells compared to neural stem cells (Figure 6F and Figure 8A,B). Sustained high levels of labile iron and ROS make N2A vulnerable to Erastin or RSL3 treatment. There may be other factors responsible for the resistance of neural stem cells, which requires further investigation. Nevertheless, overexpression of *Fth* significantly reduces ROS levels and cell death, and induces expression of GPX4 and cell viability in N2A cells (Figure 7). Most excitingly, neuroblastoma cell lines significantly express a low level of *Fth* compared to almost all other types of cancer cell lines (Figure 7E). We, therefore, conclude that the low level of *Fth* causes sensitivity of neuroblastoma to ferroptosis inducers.

Interestingly, a high level of FTH in the serum of patients is a common marker for a worse prognosis of high-risk neuroblastoma [39], which may be due to the promotion of FTH at neuroblastoma cell growth and inhibition of cell death. Our results confirmed these effects (Figure 7A,B). As previously reported, deletion of FTH leads to embryonic lethality [40], whereas deletion of FTL does not [41], probably because FTH has a more conserved and vital function, such as ferroxidase activity of FTH, which protects cells from oxidative stress, rather than FTL as a structural protein. Based on these reports and our results, it is presumed that inhibition of ferroxidase activity by FTH, combined with ferroptosis inducer in neuroblastoma, which has a poor prognosis, could be a promising therapeutic schedule.

The role of PARP1 and PARP1-mediated PARylation in ferroptosis is an intriguing research area. It has been reported that inhibition of PARP1 can increase the expression of GPX4 in mouse livers and PC12 cells [27,28]. In MDA-MB-231 cells, the use of siRNA to lower PARP1 increases NRF2 expression [29,42], while GPX4 and Nrf2 are critical regulatory factors in ferroptosis. All of this suggests that PARP1 and PAR modification may be involved in the process of ferroptosis. We did find that ferroptotic N2A cells produce high levels of PAR (Figure 8), which is the catalytic product of PARP1, suggesting that there is a crosslink between ferroptosis and the PARP1 signaling pathway. GPX4, the critical factor of ferroptosis, also participates in the necroptosis process, since necroptosis inhibitor Nec-1 can rescue cell death caused by GPX4 deficiency [43]. Chaperone-mediated autophagy (CMA) is associated with ferroptosis through Hsp90; inhibition of CMA can stabilize GPX4 and, thereby, reduce ferroptosis [44]. All of these indicate that necroptosis, autophagy, and ferroptosis may be crosslinked, among which, GPX4 plays a crucial role. PARP1-mediated cell death is known as parthanatos, and AIF is a critical member of parthanatos [31], which can be regulated by GPX4 [30,31]. These suggest that there may be a similar crosslink between ferroptosis and parthanatos. However, the use of PARP1 inhibitors—although PAR modification was wholly blocked—cell vitality, level of lipid ROS, and expression of GPX4, were not rescued.

The resistance to chemotherapy is a complex problem in the antitumor field. Most researchers have induced tumor apoptosis to achieve an antitumor effect. The discovery of ferroptosis in 2012 offers a new target for alleviating drug resistance. For example, inhibiting the *Nrf2* signaling pathway caused ferroptosis and reversed the resistance of head and neck cancer cells to cisplatin [45]. In addition to being used alone, ferroptosis inducers can be used in combination with traditional chemotherapy drugs, such as cisplatin, which can significantly increase the effect of chemotherapy on drug-resistant tumor cells [46]. In addition to drug resistance, chemotherapy drugs’ toxic/side effects are also a thorny issue. For example, STS can inhibit protein kinase C (PKC) and has a pronounced killing effect on tumors at low concentrations (~200 nM) (Figure 3E). However, its poor selectivity and high toxicity to healthy cells limit its clinical application [47,48]. In this study, neuroblastoma was very sensitive to the performance of ferroptosis inducers at low concentrations (less than 2 μM for Erastin or 1 μM for RSL3), which are almost non-toxic to healthy cells. Therefore, ferroptosis inducers have great potential in the antitumor field, opening up a new way for drug research and development.

## 4. Materials and Methods

### 4.1. The Mice

C57BL/6 pregnant mice were maintained in specific pathogen-free animal facilities of Sun Yat-Sen University (SYSU), and experiments were conducted according to licenses issued by SYSU Institutional Animal Care and Use Committee (SYSU IACUC).

### 4.2. Plasmid Construction

The empty vector pcDNA-Flag was used to construct mouse *Fth* overexpression plasmid according to the manufacturer’s instruction with One Step Cloning Kit (Vazyme, Nanjing, Jiangsu, China, C112). Two Sequences of *Nras* shRNA were cloned into the *Sac I* and *Hind III* sites of the empty vector pEGFP-U6+1. The amplified primers and shRNA oligos were shown in Appendix A, and the plasmid was confirmed by sequencing.

### 4.3. Cell Culture

N2A cells were cultured in Dulbecco’s Modified Eagle Medium (DMEM, Gibco, Carlsbad, CA, USA) containing 1% penicillin/streptomycin and 10% fetal bovine serum (FBS, Gibco, Carlsbad, CA, USA) at 37 °C and 5% CO_2_, and passed every two days.

Isolation and culture methods of primary neurons were performed as previously described [49]. The pregnant females were sacrificed, cerebral cortices from mouse embryos (E17.5) were dissected, and cortices were digested by trypsin. The tissue and filter were resuspended through a cell stainer (40 μM). Cells were seeded into poly-l-lysine-coated (Sigma, St. Louis, MO, USA) plates in plating medium (DMEM with 0.5% glucose, 1 mM sodium pyruvate, 1% penicillin/streptomycin, 10 mM HEPES, 10% FCS, 1 mM L-glutamine and 2% B-27 supplement). On the second day, we changed the medium to a neuro-basal medium (Invitrogen, Carlsbad, CA, USA) containing 2% B-27 and 0.5 mM L-glutamine, and replaced half of the medium every three days.

Primary neural stem cells were isolated from mouse embryos at 14.5 days, and then plated into T25 using as a surface, the wall of the tube that was not coated (T25 in vertical position), and cultured in the neural stem cell medium (DMEM containing 2% B27, 1% penicillin/streptomycin, 20 ng/mL bFGF and 20 ng/mL EGF). Neural stem cells formed neurospheres and were subcultured every 3 days. Before drug treatment, the neurospheres were disintegrated into single-cell suspension, and then seeded in a culture dish coated with 0.1 mg/mL poly-l-lysine for adherent growth.

### 4.4. Cell Viability Measurement (CCK-8 Assay)

Erastin (Selleck, Houston, TX, USA), RSL3 (Selleck, Houston, TX, USA), veliparib (Beyotime, Shanghai, China), rucaparib (Beyotime, Shanghai, China), liproxstatin-1 (MCE, Princeton, New Jersey, USA), staurosporine (GLPBIO, Montclair, CA, USA), and Z-VAD-FMK (Beyotime, Shanghai, China) were dissolved in DMSO and kept in −80 °C. Cells were seeded into a 96-well plate with 8000 cells/well, and DMSO, Erastin or RSL3 were added into the medium the next day. Ferroptosis inhibitor (liproxstatin-1), PARP inhibitors (veliparib and rucaparib), apoptosis inhibitor (Z-VAD-FMK) were added 2 h before Erastin and RSL3 or staurosporine. After culturing for 6 or 22 h, 10 μL CCK-8 reagent (GLPBIO, Montclair, CA, USA) was added and incubated at 37 °C and 5% CO_2_ for 2 h. The absorbance was measured at 450 nm with a microplate analyzer.

### 4.5. BrdU Labeling and Staining

Cells were seeded on slides and cultured at 37 °C and 5% CO_2_. After drug treatment, BrdU (final concentration 10 μM, Sigma, St. Louis, MO, USA) was added to the medium and incubated for 1 h. The cells were fixed with 4% paraformaldehyde, treated with 2N HCl at 37 °C for 30 min, and sealed and permeable with blocking solution (BS) (5% goat serum, 1% bovine serum albumin, 0.4% Triton X-100). BrdU antibody (Abcam, Cambridge, England, 1:200 dilution) was incubated overnight at 4 °C, followed by secondary antibody Alexa Fluor 488 (Invitrogen, Carlsbad, CA, USA, 1:200) for 1 h at room temperature and DAPI for 20 min. A mounting medium (Sigma, St. Louis, MO, USA) was used to seal the plates, and the images were observed under a fluorescence microscope (ZEISS, Oberkochen, Germany).

### 4.6. Crystal Violet Staining

After treatment with a drug-containing culture medium for 12 h, the culture medium was discarded, and the cells were washed with PBS, followed by fixation with 4% paraformaldehyde for 15 min and cleaning with PBS. Finally, we incubated with crystal violet staining solution (Beyotime, Shanghai, China) at room temperature for 10 min and rinsed with PBS. Imaging was conducted under a stereoscope.

### 4.7. Propidium Iodide (PI) Staining

For fluorescence microscope observation, the medium containing PI and Hoechst was directly added and incubated at 37 °C for 25 min to a final concentration of 5 μM and 10 μg/mL, respectively, before being used for microscope imaging.

### 4.8. Western Blotting

After drug treatment of DMSO, Erastin, RSL3, staurosporine or combination with minocycline (Sigma, St. Louis, MO, USA), veliparib, rucaparib, liproxstatin-1, the cells were lysed with NETN buffer (50 mM Tris-HCl pH 7.4, 150 mM NaCl, 1% NP40, 1 mM EDTA, plus 1 tablet of Roche complete protease inhibitor per 10 mL) to collect the total protein of the cells. After SDS-PAGE, the protein on the gel was transferred to PVDF membrane (Bio-Rad, Hercules, CA, USA), sealed with 5% milk for 1 h, and the primary antibody GPX4 (ABclonal, 1:1000 dilution), HO-1 (Proteintech, Chicago, IL, USA, 1:1000), PAR (Trevigen, Gaithersburg, MD, USA, 1:1000), Flag (Sigma, 1:3000), GAPDH (ABclonal, 1:5000), beta-actin (ABclonal, Wuhan, Hubei, China, 1:5000), or cleaved caspase-3 (Cell Signaling Technology, Boston, MA, USA, 1:1000) were incubated overnight at 4 °C. The secondary antibody with HRP was incubated at room temperature for 1 h. Proteins were visualized with the SuperSignal Chemiluminescent Substrate (Thermo Scientific, Waltham, MA, USA).

### 4.9. Lipid ROS Detection

BODIPY 581/591 C11 (Thermo Fisher, Waltham, MA, USA) was dissolved in DMSO at a concentration of 2 mM. After treatment with drugs, the cells were added with BODIPY 581/591 C11 at a final concentration of 2 μM, and then cultured at 37 °C and 5% CO_2_ for 20 min. After PBS washing, N2A cells were digested into a single cell with trypsin and analyzed by FACS. The detection channel was FITC (BODIPY 581/591C11 excitation light at 488 nm and emission light at 530 nm).

### 4.10. Measurement of Labile Iron Pool (LIP)

LIP was measured by using calcein-acetoxymethyl ester (Beyotime, Shanghai, China), a fluorescence probe, according to the manufacturer’s instructions. Briefly, cells were washed three times with PBS after trypsinization, and then stained with 0.6 μM of calcein-acetoxymethyl ester at a density of 1 × 10^6^/mL for 30 min at 37 °C in the dark. Then, the cells were washed twice with PBS and either incubated with deferiprone (100 μM) for 1 h at 37 °C or left untreated. FACS then analyzed cells with the FITC channel. The difference in the mean cellular fluorescence with and without deferiprone incubation reflected the amount of LIP.

### 4.11. RNA Isolation and PCR Analysis

For RNA extraction, 1 mL TRIzol reagent (15596026, Thermo Scientific, Waltham, MA, USA) was added to the cells in the 6-well plate, and 0.2 mL chloroform was added after oscillation. We collected the lysate and centrifuged it. The upper colorless aqueous phase was taken and 0.5 mL isopropanol was added. After centrifugation, the precipitation was collected, and we added 1 mL 75% ethanol for washing. The precipitation was dried and dissolved in RNA-free water.

The above RNA was used to synthesize the first-strand cDNA by the RevertAid First Strand cDNA Synthesis Kit (K1622, Thermo Scientific, Waltham, MA, USA), according to the manufacturer’s instruction. SYBR Green Premix (AG11701, Accurate, Changsha, Hunan, China) was used for quantitative PCR (qPCR) reactions. The primers were presented in Appendix A.

### 4.12. Analysis of Relative Expression of Fth, Gpx4, and Fsp1 in Different Cancer Cell Lines

The expression datasets of *Fth, Gpx4,* and *Fsp1* in 1378 cancer cell lines were downloaded from the public, open-access repository “Broad DepMap Portal” (https://sites.broadinstitute.org/ccle/) (Accessed on 3 August 2021). Cancer types with the number of cell lines less than 5, and engineered cancer lines, were excluded. The expressions of *Fth, Gpx4*, and *Fsp1* in 28 neuroblastoma cell lines were compared with the other 28 different types of cancer with 1354 cell lines.

### 4.13. Statistical Analysis

Data were shown as mean ± SEM. Statistical significance (*p* values) was calculated by Prism 8.0.2 (GraphPad Software, San Diego, CA, USA). The two-tailed *t*-test, one-way ANOVA test and two-way ANOVA test were used in the present study. More details of statistical analysis were described in each figure legend.

## 5. Conclusions

In summary, our study finds that N2A cells express a low level of ferritin subunit FTH, leading to a high ROS level and sensitizing RAS-proficient N2A cells to ferroptosis induction (graphical abstract). These results provide a promising therapeutic target for treating neuroblastoma and provide a new vision that the ferroptosis inducers can also specifically kill RAS-mutation-free tumor(s). We also found that there may be a potential crosslink between ferroptosis and PARP1-mediated PAR formation, which needs to be further investigated.

## Figures and Tables

**Figure 1 ijms-22-08898-f001:**
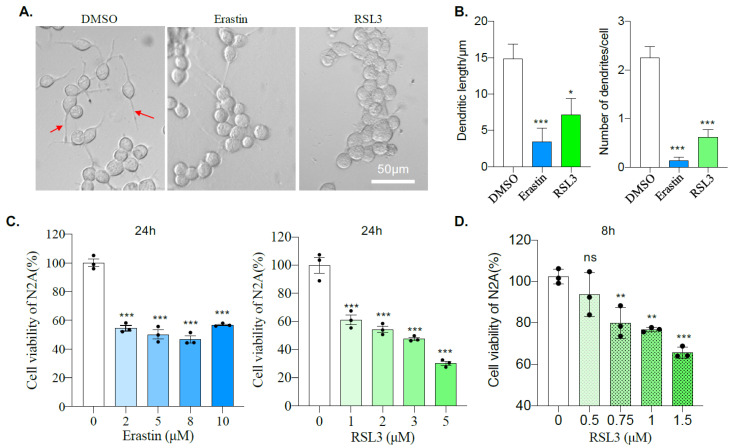
Erastin and RSL3 treatments affect the morphology and viability of N2A cells. (**A**) N2A cells were treated with DMSO, 10 μM Erastin or 5 μM RSL3, and bright-field images were taken 24 h after for morphology analysis. The red arrows indicate the dendrites. Scale bar = 50 μm. (**B**) The total neurite length and the number of primary neurites per cell from (**A**) were analyzed. *n* = 20 from three repeated experiments. *: *p* < 0.05; ***: *p* < 0.001. Neurite length was measured by Image J. (**C**) The viability of N2A cells was analyzed by CCK-8 assay at 24 h after treating with DMSO, different doses of Erastin or RSL3. (**D**) The viability of N2A cells was analyzed by CCK-8 assay at 8 h after treating with DMSO, low doses of RSL3. Data were obtained from three independent experiments. **: *p* < 0.01; ***: *p* < 0.001; ns: not significant.

**Figure 2 ijms-22-08898-f002:**
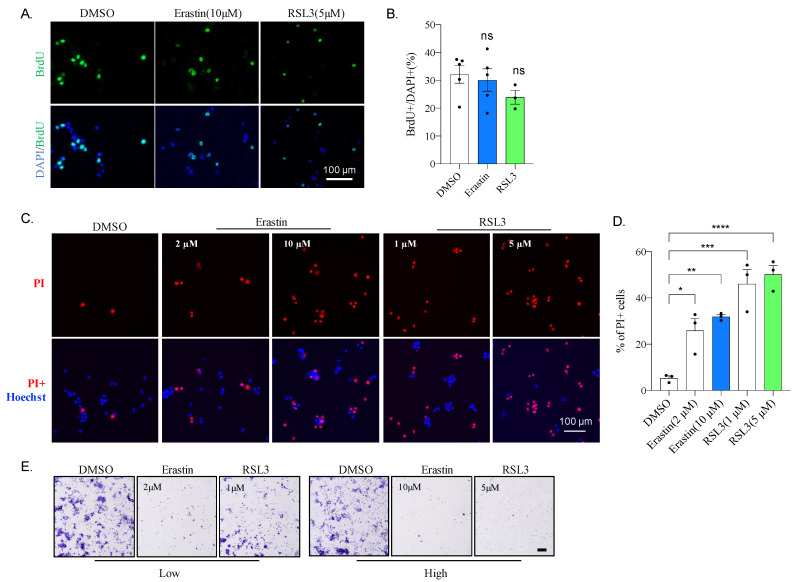
Erastin and RLS3 on proliferation and cell death in N2A cells (**A**) Representative images of BrdU staining. N2A cells were plus labeled for 1 h after treating for 23 h with DMSO, 10 μM of Erastin or 5 μM of RSL3. Cells were then fixed for BrdU antibody staining. Scale bar = 100 μm. (**B**) Quantification of (**A**), data were obtained from triplicated experiments. ns: not significant. (**C**) Fluorescent images of N2A cells with PI staining. N2A cells were treated for 12 h with Erastin or RSL3. Scale bar = 100 μm. (**D**) The quantification of (**B**) *: *p* < 0.05; **: *p* < 0.01; ***: *p* < 0.001; ****: *p* < 0.0001. (**E**). Bright-field images of N2A cells with Crystal violet staining. N2A cells were treated for 24 h with Erastin or RSL3. Scale bar = 500 μm.

**Figure 3 ijms-22-08898-f003:**
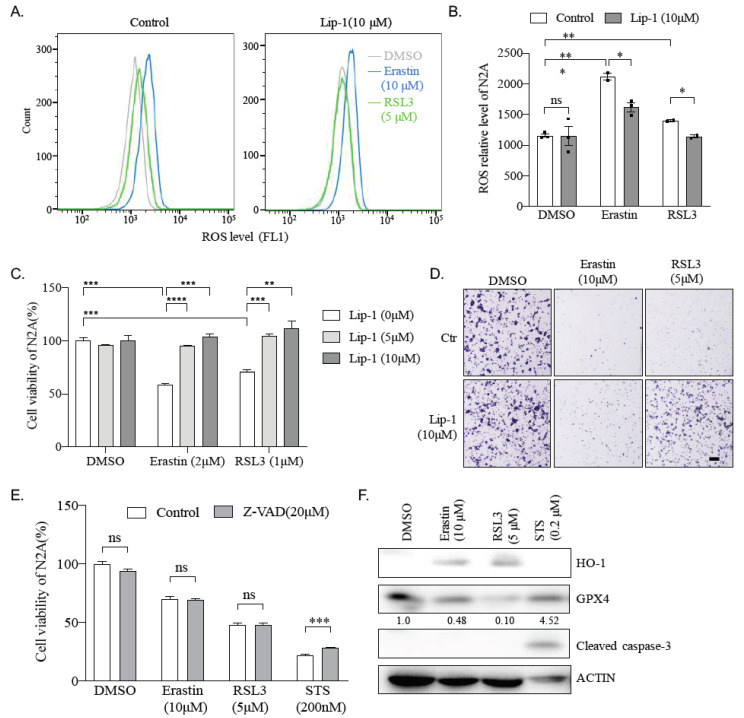
Erastin and RSL3 induce ferroptosis but not apoptosis and N2A cells. (**A**) Lipid ROS level in DMSO, Erastin, or RSL3 treated N2A cells for 24 h. Cells were treated with ferroptosis inhibitor liproxstatin-1 (Lip-1) for 2 h before incubation with DMSO, Erastin, or RSL3. Cells were then stained with BODIPY 581/591 C11, and lipid ROS level was detected by FACS. (**B**) Quantification of lipid ROS level in N2A cell from (**A**). Data was obtained from three independent experiments. *: *p* < 0.05; **: *p* < 0.01; ns: not significant. (**C**) Viability of N2A cells was measured after treating with ferroptosis inhibitor Lip-1 and indicated drugs for 24 h. Data were obtained from three independent experiments. **: *p* < 0.01; ***: *p* < 0.001; ****: *p* < 0.0001. (**D**) N2A cells were treated with DMSO, 10 μM Erastin, 5 μM RSL3 or a combination with 10 μM Lip-1, stained with crystal violet, and bright-field images were taken. Scale bar = 500 μm. (**E**) The viability of N2A cells was measured after treating with DMSO, 2 μM Erastin, 1 μM RSL3, or 200 nM apoptosis inducer staurosporine (STS), or combination with 20 μM apoptosis inhibitor Z-VAD-FMK (Z-VAD) for 24 h. ***: *p* < 0.001; ns: not significant. (**F**) N2A cells were treated with DMSO, 10 μM Erastin, 5 μM RSL3, or 200 nM STS for 24 h, and protein extraction was prepared for western blotting analysis with antibodies against HO-1, GPX4, cleaved caspase-3, and β-actin. Quantified relative expression of GPX4 to β-actin by Image J. Images represent three independent experiments.

**Figure 4 ijms-22-08898-f004:**
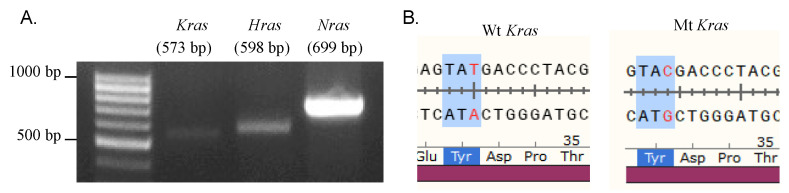
Sequencing of RAS family genes in N2A cells. (**A**). The PCR product of *Kras*, *Hras*, and *Nras* was detected by agarose. (**B**) The sequence of *Kras* gene from NCBI database (Wt *Kras*) and N2A cells (Mt *Kras*). Mt *Kras* harbors a synonymous mutation: c.96T>C (p.32Y=).

**Figure 5 ijms-22-08898-f005:**
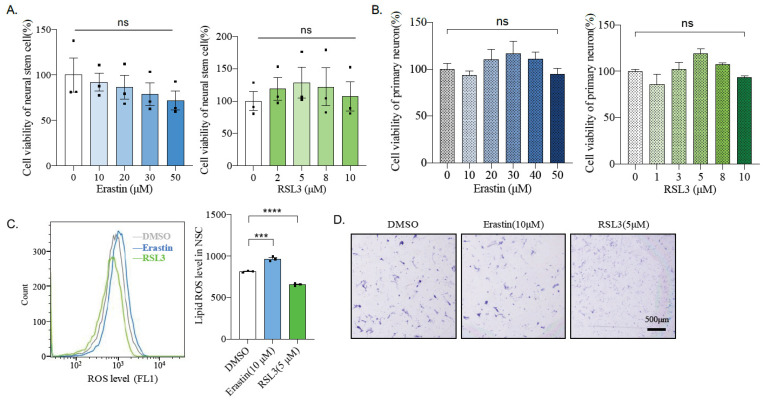
Neural stem cells and neurons are more resistant to Erastin and RLS3 treatments. (**A**) The viability of primary cortical neural stem cells was measured after treatment with indicated doses of Erastin or RSL3 for 24 h. Cells were obtained from three mice. Data were obtained from three independent experiments. ns: not significant. (**B**) The viability of primary neurons was measured after treating with indicated doses of Erastin or RSL3 for 24 h. Cells were obtained from three mice. Data were obtained from three independent experiments. ns: not significant. (**C**) Lipid ROS levels in neuronal stem cells treated with indicated drugs for 24 h and detected by FACS. Data were obtained from three independent experiments. ***: *p* < 0.001; ****: *p* < 0.0001. (**D**) Neuronal stem cells were treated with DMSO, 10 μM Erastin, 5 μM RSL3 for 24 h, stained with crystal violet, and bright-field images were taken. Scale bar = 500 μm.

**Figure 6 ijms-22-08898-f006:**
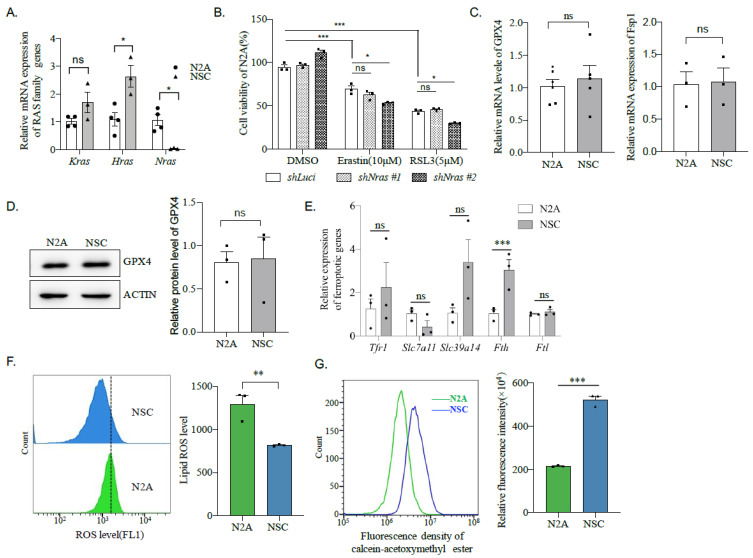
Low expression of *Fth* sensitizes N2A cells to ferroptosis inducers. (**A**) The expression of RAS family genes in N2A and neural stem cells was analyzed by qPCR. Data were obtained from the indicated number of samples. *: *p* < 0.05; ns: not significant. (**B**) Cell viability of N2A with *Nras* silencing and Erastin and RSL3 treatment. *: *p* < 0.05; ***: *p* < 0.001; ns = no significance. (**C**) The mRNA expression of *Fsp1* and *GPx4* between N2A and NSC. ns: not significant. (**D**) The protein expression of *GPx4* between N2A and NSC. ns = no significance. (**E**) The mRNA expression of *Tfr1, Slc7a11, Slc39a14, Fth,* and *Ftl* between N2A and NSC. ***: *p* < 0.001; ns: not significant. (**F**) Lipid ROS level in N2A cell and NSC detected by FACS. **: *p* < 0.01. (**G**). Labile iron pool in N2A and NSC was detected by incubation cells with 0.6 μM of calcein-acetoxymethyl ester fluorescence probe for 1h before analyzing with FACS. ***: *p* < 0.001.

**Figure 7 ijms-22-08898-f007:**
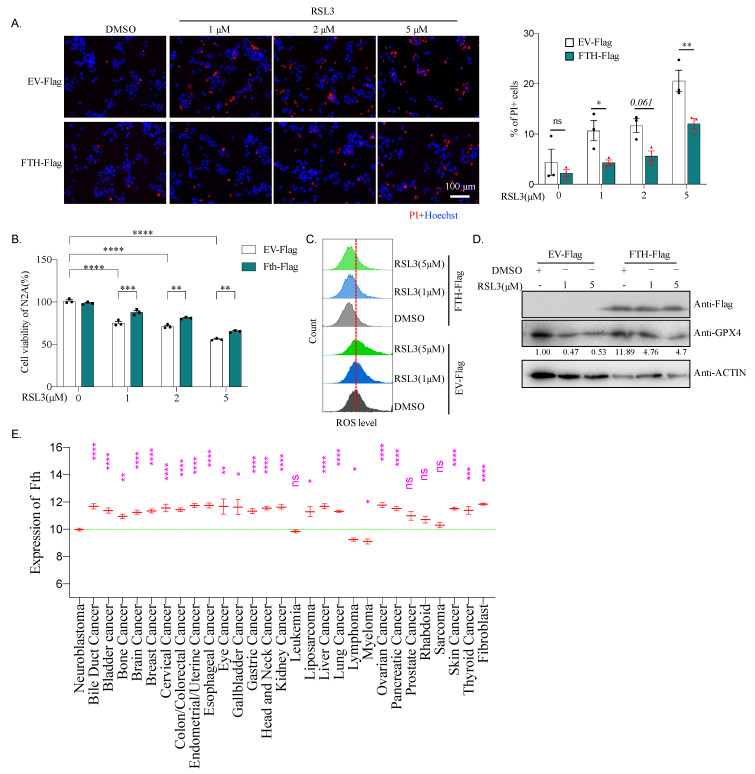
Overexpression of *Fth* rescues ferroptosis in N2A cells. (**A**) Representative images of PI/Hoechst staining of N2A cells, overexpressing pcDNA-Flag empty vector (EV-Flag) or pcDNA-Flag-FTH. The cells were treated by DMSO or 1, 2, 5 μM RSL3 for 24 h. Ratios of PI-positive cells were quantified and showed on the right. For the concentration of 2 μM RSL3, *p* = 0.061; *: *p* < 0.05; **: *p* < 0.01; ns: not significant. (**B**) Viability of N2A cells overexpress EV-Flag, or FTH-Flag was analyzed by CCK-8 assay at 24 h after treating with DMSO, or 1, 2, 5 μM RSL3. **: *p* < 0.01; ***: *p* < 0.001; ****: *p* < 0.0001. (**C**) Lipid ROS level in N2A cells of overexpressing EV-Flag or FTH-Flag treated by DMSO, or 1 μM, 5 μM RSL3 for 24 h. (**D**) Western blotting analysis of N2A cells overexpressing EV-Flag or FTH-Flag treated by DMSO, or 1 μM, 5 μM RSL3 for 24 h. Quantified relative expression of GPX4 to β-actin by Image J. (**E**) Relative expression of *Fth* in neuroblastoma and other 28 cancers. Data were extracted from the CCLE database (https://sites.broadinstitute.org/ccle/) (Accessed on 3 August 2021), a public, open access repository. *: *p* < 0.05; **: *p* < 0.01; ***: *p* < 0.001; ****: *p* < 0.0001; ns: not significant.

**Figure 8 ijms-22-08898-f008:**
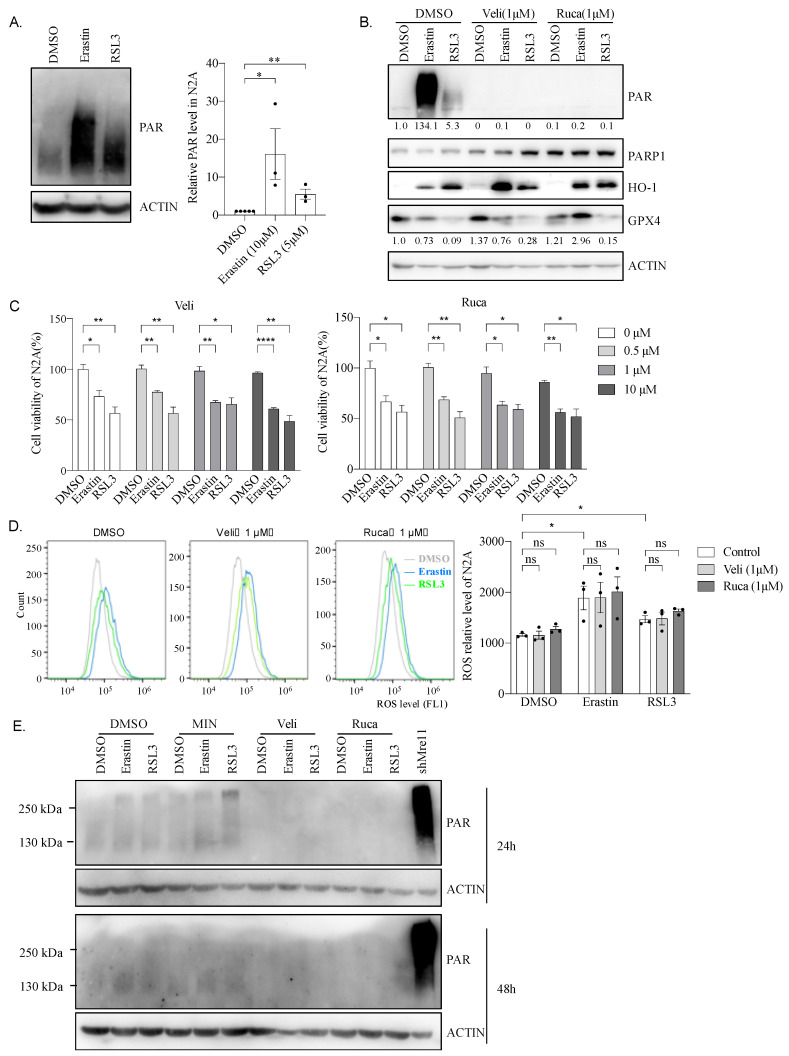
Erastin and RLS3 induce cell death independent of hyper PARP activity. (**A**) Erastin and RLS3 induce a high level of PAR in N2A for 24 h. Representative images of western blotting were shown on the left. Quantified relative expression of PAR to β-actin by Image J and the value was shown on the right. Data were obtained from three independent experiments. *: *p* < 0.05; **: *p* < 0.01. (**B**) Western blotting analysis of PAR, PARP, HO-1, and GPX4. N2A cell treated with DMSO, 10 μM Erastin or 5 μM RSL3 for 24 h, or combined with 1 μM PARP inhibitor of veliparib (Veli) or rucaparib (Ruca) 2 h before with the indicated drugs. Protein extraction was prepared for Western Blotting analysis with indicated. Quantified relative expression of PAR or GPX4 to β-actin by Image J. Images represent three independent experiments. (**C**) Viability of N2A cells was measured after treating with indicated doses of PARP1 inhibitors Veli and Ruca together with ferroptosis inducers (10 μM Erastin or 5 μM RSL3) for 24 h. Data were obtained from three independent experiments. *: *p* < 0.05; **: *p* < 0.01; ****: *p* < 0.0001. (**D**) Lipid ROS level in N2A cells treated with indicated drugs for 24 h and detected by FACS. Data were obtained from three independent experiments. *: *p* < 0.05; ns: not significant. (**E**). Western blotting analysis of PAR in primary neurons. Neurons treated with DMSO, 50 μM Erastin or 5 μM RSL3 for 24 h (up) or 48 h (down). Neuron was also treated by a PARP inhibitor of 0.1 μM MIN (minocycline), 1 μM Veli or 1 μM Ruca for 12 h before being treated with ferroptosis inducers. Protein extraction was prepared for Western Blotting analysis with indicated. The last lane of sh*Mre11* was used as a positive control for PAR.

**Table 1 ijms-22-08898-t001:** RAS Sequencing results of N2A cells.

Gene	Sequencing Results
*Kras*	Synonymous mutations, c.T96>C
*Hras*	No mutation
*Nras*	No mutation

## Data Availability

The data presented in this study are available on request from the corresponding authors.

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
