# Peer review of "A Shortage of FTH Induces ROS and Sensitizes RAS-Proficient Neuroblastoma N2A Cells to Ferroptosis"

_ijms, 2021, doi:10.3390/ijms22168898_

Round 1
Reviewer 1 Report
In this manuscript, Lu et al investigated the role of ferroptosis in neuroblastoma N2A cells and suggest that low expression of FTH in N2A cells compared to neuroprogenitor cells contributes to ferroptosis. Since neuroblastoma is one of the most incurable tumor, it is important to suggest ferroptosis as a new therapeutic strategy. Most experiments are well designed and conducted. However, only one cell line was imployed while several studies have already suggest ferroptosis in neuroblastoma. In addition, suggested mechanism are very common in other cancer and normal tissues. There are several issues as follow:
- Although relative low expression of FTH in N2A cells compared to NSC is the key observation in this study, there is no evidence whether that such a difference directly affect basal ROS levels and RSL3- or erastin-induced ferroptosis. At least, the author should show labil iron pool in both cells. In addition, it is also curious how overexpression of FTH facilitates proliferation in Fig. 6G. If relative viability were calculated, FTH-overexpressing cells seem more sensitive to erastin and RSL3, arguing that RSL3 and erastin can bypass FTH. Please do not cut the Y axis in Fig. 6G.
- Related to above question, cancer cells and normal cells are completely different in their gene expression and metabolim, which can result in differential sensitivity to ferroptosis in addition to FTH. The authors simply can analyze whether neuroblastoma cells express low levels of FTH compared to other type of cancer cells from CCLE database (https://sites.broadinstitute.org/ccle/).
- 5C, there are no significant increase in lipid peroxidation. This might be due to the experimental condition. Lipid peroxidation upon RSL3 is normally very fast and transient, normally BODIPY signal is the highest at 1-4 h treatment of RSL3, then the signals disappears. In erastin treatment, lipid peroxidation is visuable a few hours before cells die.
- Indeed, 2-5 uM RSL3 to induce cancer cell death is relatively higher than in other cancer cells, suggesting neuroblastoma is not highly sensitive to ferroptosis.
- 200 nm – 500 nm of Liproxstatin-1 is sufficient to inhibit ferroptosis in most study while the author used at 5 – 10 uM.
- In Fig. 3F, it is uncertain whether GPX4 is indeed reduced by erastin and RSL3 because there are huge decrease in actin levels because the authors used lysates treated for 24 h which 50% cells already died
Author Response
In this manuscript, Lu et al investigated the role of ferroptosis in neuroblastoma N2A cells and suggest that low expression of FTH in N2A cells compared to neuroprogenitor cells contributes to ferroptosis. Since neuroblastoma is one of the most incurable tumor, it is important to suggest ferroptosis as a new therapeutic strategy. Most experiments are well designed and conducted. However, only one cell line was imployed while several studies have already suggest ferroptosis in neuroblastoma. In addition, suggested mechanism are very common in other cancer and normal tissues. There are several issues as follow:
Response 1:
We appreciated this reviewer for her/his positive view of our study and for his/her constructive comments on the manuscript. Indeed, several studies have already reported the function of ferroptosis in neuroblastoma. However, it is still not clear that why neuroblastoma sensitizes to ferroptosis induction. Our study takes advantage of primary cells from mice to investigate the function of ferroptosis in neuroblastoma by comparing primary cells and mouse-originated neuroblastoma N2A cells, from which we identify that FTH1 could be one of the main factors that sensitize RAS-proficient neuroblastoma to ferroptosis inducers. In addition to N2A cells, we use the primary neuroprogenitor (NSC) or neurons isolated from different animals in our study. Following the comments from this reviewer, we have conducted experiments to confirm some of our data in another cell line SH-SY5Y, a human neuroblastoma cell line, which also shows a sensitivity to ferroptosis. And now, we included these data in supplemental Fig S4 and described the result in main text line 617~618 in the revised manuscript. FTH1 is a common regulator of ferroptosis in other cancer types, especially in Ras-mutant cancers (P MID: 18355723) , our study, for the first time, investigates its role in Ras-proficient neuroblastma. Moreover, following the suggestion from this reviewer ( Point 2), we compared the expression of FTH1 between neuroblastma cell lines and other cancer cell types. Interestingly, we found that neuroblastoma cell lines express a significantly lower amount of FTH1 than most other cancer cells, suggesting that FTH1 could be a specific target for ferroptosis inducers in neuroblastoma. We included this data in our revised manuscript in Fig 7E and Fig S3 (see also point 2 from this reviewer), and results were described or discussed in the main text of the revised manuscript (see line25~26, line 604~617, and line 862~864).
- Although relative low expression of FTH in N2A cells compared to NSC is the key observation in this study, there is no evidence whether that such a difference directly affect basal ROS levels and RSL3- or erastin-induced ferroptosis. At least, the author should show labil iron pool in both cells. In addition, it is also curious how overexpression of FTH facilitates proliferation in Fig. 6G. If relative viability were calculated, FTH-overexpressing cells seem more sensitive to erastin and RSL3, arguing that RSL3 and erastin can bypass FTH. Please do not cut the Y axis in Fig. 6G.
Response 2:
We thank this reviewer for her/his very constructive suggestions. Following her/his suggestions, we analyzed the labile Iron pool(LIP) by using a calcein-acetoxymethyl ester fluorescence probe. Indeed, we found that N2A cells show a lower level of calcein-acetoxymethyl ester fluorescence, which suggests a high LIP inside the N2A cells compared to NSC. This data is now included in the revised manuscript as Fig 6G and described in line 564~569. This data correlated well with the factor that NSC cells harbor a lower level of ROS (Fig 6F). To further examine the effect of Fth on the ferroptosis, we overexpressed a Flag-tagged Fth instead of GFP-tagged on in the original version in N2A cells. Our results show that overexpression of FTH significantly decreases the PI+ cells, increases cell viability, reduces ROS levels, and induces expression of GPX4 in N2A. All these suggest the Fth1 plays an important role in ferroptosis neuroblastoma. There data are now included in Figure 7 and described in line 596~601 in the revised manuscript.
The Fig.6G of the old version manuscript, the viability of N2A cells overexpressing FTH was higher than the control group, but this may not be necessary an effect of proliferation. It could block cell death and therefore increase of cell viability of the pool. After calculating the relative viability(data not shown), GFP-tagged FTH seems to play an ignoble effect on cells viability, which may due to the size of the tag is too large (27kd) and therefore alter the function of Fth. We now repeat all our experiments with a Flag-tagged vector. And the Fig 6G result was replaced by Fig 7B in the revised manuscript.
- Related to above question, cancer cells and normal cells are completely different in their gene expression and metabolim, which can result in differential sensitivity to ferroptosis in addition to FTH. The authors simply can analyze whether neuroblastoma cells express low levels of FTH compared to other type of cancer cells from CCLE database (https://sites.broadinstitute.org/ccle/).
Response 3:
We thank this viewer for her/his very constructive and vital comment. Following her/his comment, we download the expression data of FTH in 1378 cancer cell lines from the “Broad DepMap Portal" which is the most recently processed and up-to-date CCLE datasets. After excluding 16 cell lines, including 1x Adrenal Cancer,1x Embryonal Cancer,1x Teratoma, and 13x Engineered cancer lines, expression of FTH1 in 1362 cancer cell lines belong to 29 types of tumors were analyzed. Strikingly, except two types of cancer cell lines ( lymphoma and myeloma) express significant low levels of FTH1 and four types of cancer cell lines (Leukemia, Sarcoma, Rhabdood, and Prostate cancer) express a similar or slight high level of FTH1, all other 22 types of cancer cell lines express a significant higher level of Fth1 compared to neuroblastoma. We also analyzed the expression of GPX4 and FSP1 in these datasets. We found the neuroblastoma expresses a similar amount of GPX4 as most other cancer cell lines. The expression of Fsp1 is also similar to some other cancer cell lines. These data suggest that a low level of Fth is a crucial marker of neuroblastoma, which could sensitize Neuroblastoma to ferroptosis. We have now included this essential data in Fig 7F and Fig S3 in the revised manuscript. We also included and described this vital information and data in the abstract (see line 25~26) and the main text (see line 604~617, and line 862~864) in the revised manuscript.
- 5C, there are no significant increase in lipid peroxidation. This might be due to the experimental condition. Lipid peroxidation upon RSL3 is normally very fast and transient, normally BODIPY signal is the highest at 1-4 h treatment of RSL3, then the signals disappears. In erastin treatment, lipid peroxidation is visuable a few hours before cells die.
Response 4:
We did analyze ROS levels in early time points and found the neither Erastin nor RSL3 induces ROS 4hr after drug treatments in NSC. And the slight induction of ROS was evidenced 24h after Erastin but not RSL3 treatment. This phenomenon is different from N2A cells in which ROS induction was detected in both Erastin and RSL3 24hr after (See Fig3A-B).To be consistent with treatment with N2A cells, we presented data from 24hr for NSC. We have now included the result from 4hr after treatment in supplemental Fig S2 and in line 508~509 in the main text of the revised manuscript. Therefore, we believe that NSC is resistant to Erastin, especially to RSL3 induced ROS, compared to N2A cells.
- Indeed, 2-5 uM RSL3 to induce cancer cell death is relatively higher than in other cancer cells, suggesting neuroblastoma is not highly sensitive to ferroptosis.
Response 5:
We agree with this reviewer that other cancer cells sensitize to RSL3 in a lower concentration than 2~5uM, especially for the RAS-mutated cancer cells (PMID: 18355723 ). However, 2~5uM of RSL3 are typical concentrations used for other cancer cell types( PMID: 24439385). Following the comment from this reviewer, we titter the concentration further to test how N2A cells are sensitive to RSL3 and found the 0.5~0.75uM is sufficient significantly to induce cell death and inhibit the viability of N2A cells. This piece of data is now included in Fig S1a-b and Fig 1D in the revised manuscript. With the tested concentrations, N2A cells are susceptible to ferroptosis compared to primary NSC or neurons. Nevertheless, to state the result more precisely, we deleted “highly” in line 337 of the original version in the main text.
- 200 nm – 500 nm of Liproxstatin-1 is sufficient to inhibit ferroptosis in most study while the author used at 5 – 10 uM.
Response 6:
We started with 2uM of Lip-1 inhibitor to block ferroptosis at the beginning of the project. However, the efficiency is not as good as expected(see below one of these results). Therefore we increase it to 5~10uM. Follow the comment from this reviewer, we carried out experiments to rescue RLS3 induced cell death by 200 nM and 500 nM of Liproxstatin-1 treatment, which is now included in Fig S1c and in line 427~428 of the revised manuscript. Nevertheless, 5 -- 10 μM of Liproxstatin-1 did not show any toxicity to cells(Fig 3C-D), which, we believe, do not affect the results of this study.
- In Fig. 3F, it is uncertain whether GPX4 is indeed reduced by erastin and RSL3 because there are huge decrease in actin levels because the authors used lysates treated for 24 h which 50% cells already died
Response 7:
We agree with this reviewer that the level of ACTIN was also decreased after Erastin or RSL3 treatments in Fig 3F, where a huge decrease of ACTIN was found in apoptosis inducer –STS treatment. However, the relative level of GPX4, which was calculated by dividing GPX4 level by ATCIN level, was still decreased after Erastin or RSL3 treatments. The representative blot of triplicated results and the value of the relative level of GPX4 (after re-calculation) were updated now in Fig 3F. Second, the samples for Fig 3F were prepared from the leftover cells after drug treatments. And then, the same amount of protein extraction was loaded after measuring the protein concentration. Therefore, we believed that the dead cells would not affect the results. Moreover, decreased expression of GPX4 was evidenced in other experimental setting ( see Fig 8B in the revised manuscript). Nevertheless, we performed WB analysis again 6hr after treating the cell with RSL3, which also showed a decrease in GPX4 expression. This data is now included in Fig 7D in the revised manuscript. Taking all these together, we conclude that Erastin or RSL3 treatment reduces the expression of GPX4.
Reviewer 2 Report
This study would be of interest to researchers studying cancer and ferroptosis, and a number of mechanistic investigations have been done to both determine the responsible cause (FTH) and rule out other possibilities.
There are 2 significant aspects which detract from this work.
1) There are far too many grammatical or formatting errors. Some quick ones are repeated lines 120-16 with lines 113-110 and (presumably) an errant carriage return in line 249 which leads to line 259.
2) The size of the font for many figures is too small- which may reflect the formatting of figures for IJMS. Examples are Figure 1, 5, 6, and 7. Such small font makes it hard to appreciate and highlight the hard work and significant findings.
Methods: I did not see Minocycline mentioned in the Methods.
Author Response
This study would be of interest to researchers studying cancer and ferroptosis, and a number of mechanistic investigations have been done to both determine the responsible cause (FTH) and rule out other possibilities.
We thank the reviewer for her/his very positive evaluation of our manuscript.
There are 2 significant aspects which detract from this work.
1) There are far too many grammatical or formatting errors. Some quick ones are repeated lines 120-16 with lines 113-110 and (presumably) an errant carriage return in line 249 which leads to line 259.
We apologize for the repeated text, which is due to a mistake during the transfer of the normal word document to the template of IJMS. The errant carriage return in line 249 was removed. We also performed extensive English editing and corrected more than 200 grammatical or formatting errors.
2) The size of the font for many figures is too small- which may reflect the formatting of figures for IJMS. Examples are Figure 1, 5, 6, and 7. Such small font makes it hard to appreciate and highlight the hard work and significant findings.
We have changed the font size to "12" for all our figures, including figure1 5, 6, and 7.
Methods: I did not see Minocycline mentioned in the Methods.
We have included the protocol for Minocycline in the section of “Material and Methods”, see also in line 1227~1228 in the revised manuscript.
Round 2
Reviewer 1 Report
Although I am not fully agree with some explanation (for example, the authors could not detect lipid peroxidation in ealry time point and
,but only detect slight increase in ROS when cell died, and high concentation of liproxstatin is required to fully inhibit ferroptosis), the authors sincerely responsed and provide their interpretation